# Inhibition by Nitrogen Addition of Moss-Mediated CH_4_ Uptake and CO_2_ Emission Under a Well-Drained Temperate Forest, Northeastern China

**DOI:** 10.3390/plants15010166

**Published:** 2026-01-05

**Authors:** Xingkai Xu, Jin Yue, Weiguo Cheng, Yuhua Kong, Shuirong Tang, Dmitriy Khoroshaev, Vladimir Shanin

**Affiliations:** 1State Key Laboratory of Atmospheric Environment and Extreme Meteorology, Institute of Atmospheric Physics, Chinese Academy of Sciences, Beijing 100029, China; yuejin@mail.iap.ac.cn; 2Department of Atmospheric Chemistry and Environmental Science, College of Earth and Planetary Sciences, University of Chinese Academy of Sciences, Beijing 100049, China; 3Faculty of Agriculture, Yamagata University, Tsuruoka 997-8555, Japan; cheng@tds1.tr.yamagata-u.ac.jp; 4College of Forestry, Henan Agricultural University, Zhengzhou 450046, China; y.kong@henau.edu.cn; 5School of Breeding and Multiplication (Sanya Institute of Breeding and Multiplication), Hainan University, Sanya 572025, China; srtang@hainanu.edu.cn; 6Institute of Physicochemical and Biological Problems in Soil Science, Russian Academy of Sciences, Pushchino 142290, Russia; d.khoroshaev@pbcras.ru (D.K.); shaninvn@gmail.com (V.S.)

**Keywords:** greenhouse gas, long-term experiment, Moss, N deposition, interactive effect, forest ecosystem

## Abstract

Nitrogen (N) deposition poses a multi-pronged threat to the carbon (C)-regulating services of moss understories. For forest C-cycle modeling under increasing N deposition, failure to mechanistically incorporate the moss-mediated processes risks severely overestimating the C sink potential of global forests. To explore whether and how N input affects the moss-mediated CH_4_ and carbon dioxide (CO_2_) fluxes, a five-year field measurement was performed in the N manipulation experimental plots treated with 22.5 and 45 kg N ha^−1^ yr^−1^ as ammonium chloride for nine years under a well-drained temperate forest in northeastern China. In the presence of mosses, the average annual CH_4_ uptake and CO_2_ emission in all N-treated plots ranged from 0.96 to 1.48 kg C-CH_4_ ha^−1^ yr^−1^ and from 4.04 to 4.41 Mg C-CO_2_ ha^−1^ yr^−1^, respectively, with a minimum in the high-N-treated plots, which were smaller than those in the control (1.29–1.83 kg C-CH_4_ ha^−1^ yr^−1^ and 4.82–6.51 Mg C-CO_2_ ha^−1^ yr^−1^). However, no significant differences in annual cumulative CO_2_ and CH_4_ fluxes across all treatments occurred without moss cover. Based on the differences in C fluxes with and without mosses, the average annual moss-mediated CH_4_ uptake and CO_2_ emission in the control were 0.77 kg C-CH_4_ ha^−1^ yr^−1^ and 2.40 Mg C-CO_2_ ha^−1^ yr^−1^, respectively, which were larger than those in the two N treatments. The N effects on annual moss-mediated C fluxes varied with annual meteorological conditions. Soil pH, available N and C contents, and microbial activity inferred from δ^13^C shifts in respired CO_2_ were identified as the main driving factors controlling the moss-mediated CH_4_ and CO_2_ fluxes. The results highlighted that this inhibitory effect of increasing N deposition on moss-mediated C fluxes in the context of climate change should be reasonably taken into account in model studies to accurately predict C fluxes under well-drained forest ecosystems.

## 1. Introduction

Mosses are widely distributed under forest stands from tropical to boreal zones [1,2,3], and the photosynthetically fixed carbon (C) by the forest stand moss species can partly contribute to total primary productivity of the tree canopy [3,4], depending on the diversity and composition of moss species, climate, hydrology, geography, and human activity [2]. As a nitrogen (N)-fixing plant that produces a symbiotic interaction with N-fixing cyanobacteria [5,6,7], bryophytes have the potential to improve soil fertility under well-drained forests while simultaneously absorbing carbon dioxide (CO_2_) and methane (CH_4_) from the atmosphere [8,9,10,11]. Hence, bryophytes contribute to soil C cycle, CO_2_ emission and CH_4_ uptake at the soil-atmospheric interface under well-drained forest stands, via the moss’s photosynthesis, respiration, litter input, and the changes in moss-mediated soil properties (e.g., soil moisture, labile C, and nutrient status) [1,8,10,11,12].

Moss biomass contains energy-rich compounds such as cellulose, hemicellulose, starch, and lipids. In tropical forest ecosystems, the net annual release of soluble sugars from epiphytic bryophytes was estimated at 1.22 Mg C ha^−1^ [1], and the leakage of readily soluble C from *Hylocomium splendens* feather-moss mats in a subalpine forest in Canada reached up to 15 kg C ha^−1^ during rehydration [12]. The leakage of photosynthates produced by the mosses would be a suitable C source for heterotrophic microorganisms active in CH_4_ consumption and CO_2_ production [7,8]. Regarding the roles of mosses in regulating soil C cycle and C fluxes in terrestrial ecosystems, earlier studies have mostly focused on the symbiotic interaction between methanotrophs and aquatic bryophytes in regulating C cycle, namely the CO_2_ formed during CH_4_ oxidation for *Sphagnum* biomass and the release of oxygen upon photosynthesis of submerged *Sphagnum* mosses for methanotrophs [6,13,14,15]. This dual benefit highlights the vital role of aquatic mosses in enhancing terrestrial C sink and the mitigation of climate change. Upon the photosynthesis of mosses, the translocation of photosynthates (e.g., carbohydrates) and oxygen released from moss blankets may promote the activity of soil heterotrophs including methanotroph, resulting in an increase in CH_4_ uptake and CO_2_ emission under well-drained, moss-covered forest floors. Compared with the plots with the removal of bryophytes, an increase in dissolved organic C (DOC) and microbial biomass C contents in the soils at 5 cm depth with bryophytes was reported by Sun et al. [8] in subalpine forest and shrubland ecosystems. The increased C availability and the resulting high heterotrophic activity would support an increase in CO_2_ emission and CH_4_ uptake under the bryophyte-covered forest floors [8,10]. While previous studies have reported a positive relationship between soil DOC content and CH_4_ uptake in forest soils under laboratory and field conditions [16,17], there has been still a controversy about the variations in CH_4_ uptake by forest soils when labile C (e.g., glucose) is added alone and together with exogenous N input [18,19,20]. Presently, there is a lack of information regarding how soil C and N availability affects CH_4_ uptake and CO_2_ emission under moss-covered forest floors [8,10]. Moreover, the relationship between soil C and N availability and the moss-mediated CO_2_ and CH_4_ fluxes under well-drained forest stands remains unclear, especially in the context of long-term enhanced N input in northeastern China.

Since the industrial revolution, anthropogenic atmospheric N deposition has increased up to 105 million tons of N each year around the world, causing important impacts on terrestrial ecosystems [21]. Among these impacts are included negative effects of increased N input on the characteristics of moss species like biological N fixation [5,22] and moss growth [23,24,25,26,27], thus affecting the moss-mediated CO_2_ and CH_4_ fluxes. Furthermore, the addition of exogenous N can influence soil C availability via regulating the moss-mediated C input [1,12] and microbial decomposition of soil organic matter [28,29], which may be closely associated with soil CH_4_ and CO_2_ fluxes in forest ecosystems [8,10,17]. Besides the increased N levels, the reduction in soil pH caused by N input [30,31] and the decreased C availability caused by the inhibition of moss growth would thus have important impacts on moss-mediated CH_4_ and CO_2_ fluxes under well-drained forests. Unfortunately, the sensitivity of moss-mediated CO_2_ and CH_4_ fluxes to chronic N input is not currently well understood; for example, few studies showed that N input and moss removal did not produce a synergistically inhibiting impact on CH_4_ fluxes from a boreal peatland, although both had time-dependent negative effects on the CH_4_ fluxes [32]. To date, there have been no studies reporting the effects of chronic N addition on the moss-mediated CO_2_ and CH_4_ fluxes under forest stands and their responses to varying annual meteorological variables (e.g., precipitation, air temperature, and the duration of ground snowpack). Furthermore, there is currently a research gap in this area, specifically with regard to the mechanisms involving interactive effects of mosses and N deposition on CO_2_ and CH_4_ fluxes at the soil-atmospheric interface under well-drained forests.

An increase in soil CH_4_ uptake potential after the removal of moss layers was observed in a *Cunninghamia lanceolata* subtropical plantation forest [9]. Although moss removal did not significantly affect soil CH_4_ flux, a relatively small cumulative soil CH_4_ uptake after a 4-year moss removal occurred in a subalpine coniferous forest [10]. A similar reduction in CO_2_ emission after the removal of mosses was reported in subalpine forest and shrubland ecosystems [8]. However, the moss removal did not cause a significant change in the soil CH_4_ flux under a boreal fire chronosequence in northern Europe [33]. Based on laboratory mesocosm experiments, the short-term addition of N enhanced CH_4_ oxidation in *Sphagnum* mosses in the light rather than in the dark, highlighting the function of photosynthesis in the overall CH_4_ cycle in the *Sphagnum*-dominated peatlands [7]. One reason would be the leakage of photosynthates upon N addition as additional C sources for methanotrophs [7,14]. Probably, the moss-mediated CH_4_ fluxes under well-drained forests can be variable, depending on site-specific soil properties, the characteristics of moss species, understory light and hydrothermal conditions. Although many previous studies have revealed negative effects of N input on the growth of mosses [23,24,25,26,27], the N effects can depend on the amount and duration of added N, moss species, and site-specific meteorological conditions [24,34]. Unfortunately, the effects and driving factors of chronic N input on moss-mediated CH_4_ uptake and CO_2_ emission under well-drained forests remain unknown, especially under multi-year changing meteorological conditions.

As one of main components of photosynthates released by bryophytes, carbohydrates are generally enriched in ^13^C relative to lignin and other woody tissues by 3 to 6‰ [35,36,37]. The input of labile C components like carbohydrates increased the activity of soil heterotrophy and microbial degradation of soil native organic matter [38], resulting in more depleted δ^13^C values of soil-respired CO_2_. Furthermore, the reduction in soil pH [31] and easily decomposable soil organic C pools in forest stands [39] upon chronic N application may hamper decreases in the δ^13^C values of CO_2_ derived from soil microbial respiration [40]. Together with negative effects of increased N input on the growth of mosses [24,25,26,27], it can be hypothesized that shifts in δ^13^C and δ^18^O values of CO_2_ released from N-treated and non-treated plots with and without moss cover [41] can partly reflex soil C turnover and their relationships with the moss-mediated CO_2_ and CH_4_ fluxes under well-drained forest floors, which can provide new insights into the effect of increased N input on the moss-mediated CO_2_ and CH_4_ fluxes.

In northeastern China, the white birch-dominated deciduous forest belongs to an early stage of forest succession, which tends to prefer N supply, and mosses are distributed under the forest floors due to photophilic tree species and low forest canopy. In the context of increasing atmospheric N deposition, we hypothesize that long-term N enrichment can suppress moss-mediated CH_4_ uptake and CO_2_ emission via its negative impacts on moss growth and soil microbial processes under well-drained forests, and that the extent to which increased N input inhibits moss-mediated CH_4_ uptake and CO_2_ emission depends on meteorological conditions. The objectives of this study are to explore the effects of chronic N input on the moss-mediated CO_2_ and CH_4_ fluxes under the well-drained temperate deciduous forest, northeastern China based on a 5-year field measurement, and the key influencing factors, by taking into account the changes in soil properties such as C availability, pH, and δ^13^C and δ^18^O values of respired CO_2_ as well as meteorological factors. The results would provide a foundation for understanding and modeling the impacts of increasing N deposition on moss-mediated C fluxes and soil C sink in forest ecosystems, especially in the context of climate change.

## 2. Materials and Methods

### 2.1. Layout of Long-Term N Manipulation Experiment

This field study was carried out in a secondary temperate forest stand (42°24′9” N and 128°5′45” E), with dominated trees of white birch (*Betula platyphylla* Suk.) and mountain poplar (*Populus davidiana* Dode), at the foot of Changbai mountains, northeastern China. The dominated moss species grown mostly beneath the forest floor are *Haplocladium microphyllum* (Hedw.) Broth., *Heterophyllium affine* Fleisch., and *Pylaisia polyantha* (Hedw.) Schimp. The annual average air temperature of the study region is 4.1 °C, and annual average precipitation is 770 mm from 2004 to 2024, with more than 80% precipitation from May to October annually. The forest soil belongs to Andosols [42], with a <2 cm organic horizon and approximately 10 cm depth of A layer. Main soil properties, like soil organic matter content, pH value, texture, and bulk density were described by previous references [16,43].

A series of long-term N simulation experiments has been carried out since October 2010 to determine soil greenhouse gas (GHG) fluxes under the forest stand. The long-term N manipulation experiment contained the input of ammonium chloride at two doses (22.5 and 45 kg N ha^−1^ yr^−1^, namely low N and high N, respectively) and the control. The amount of N input at low and high doses approximated twice and four times the annual dissolved total N (DTN) input via natural forest throughfall, respectively. In the N manipulation experiment, the impacts of litter removal and its combination with N input on soil GHG fluxes were studied. Each experimental treatment included four independent plots with an area of 2 m by 8 m, and the amount of N as ammonium chloride was equally added to the N-treated plots monthly using a back sprayer in May till October annually, and an equal amount of water (10 L) was sprayed on the control plots. A detailed description of the long-term N manipulation experiment was described in Xu et al. [41]. To explore whether and how the long-term N addition affects the moss-mediated CO_2_ and CH_4_ fluxes under the temperate forest stand, following 9 years of N input, CH_4_ and CO_2_ fluxes were measured at intervals from March to November in 2019–2024 in the litter-removal experimental plots, by using soil collars without and with moss cover. The distribution of moss species across all experimental plots was uniform, and the coverage of mosses in long-term N-treated experimental plots, especially in the high-N-treated plots, showed an obvious reduction. To measure interannual changes in DTN fluxes via natural forest throughfall, 16 self-made throughfall collections have been permanently placed in the field across all experimental plots since October 2010 to collect throughfall samples at regular intervals from May to October annually. All throughfall samples were filtered via cellulose-acetate membrane (0.45 μm pore size) and frozen at −18 °C prior to the measurement of properties such as DTN. In autumn seasons of 2019, 2021 and 2024, two composed soil samples in each experimental plot with mosses were collected using a soil auger (3.1 cm in diameter and 10 cm in height), sieved by the 2 mm mesh to remove debris and small stones, and were kept in the dark at 5 °C prior to the analysis of soil attributes like moisture, pH, microbial biomass C and N, DOC, and DTN. During the period from March 2019 to November 2024, air temperature at approximately 1.0 m above the ground under the forest stand was recorded using an ONSET HOBO temperature logger [44] to obtain average annual air temperature and average non-growing season air temperature (from 1 November to 30 April), respectively. Data on daily precipitation were obtained from the National Research Station of Changbai Mountain Forest Ecosystems near this forest. The duration of ground snowpack under the forest each year was manually recorded in days.

### 2.2. Measurements of CO_2_ and CH_4_ Fluxes as Well as δ^13^C and δ^18^O Values of Respired CO_2_

In each litter-removal plot, two pairs of soil collars were inserted into the soil to a depth of about 7 cm and put perpetually in the field without and with moss cover, to determine C fluxes at the soil-atmospheric interface and isotopic values of respired CO_2_ at regular intervals during the snow cover-free periods annually. There were eight pairs of soil collars in total in each treatment. Since spring 2019, CO_2_ and CH_4_ fluxes from moss-covered and moss-free soil collars were measured on non-rainy days at regular intervals during the snow cover-free periods annually, using a portable greenhouse gas analyzer (915-0011, Los Research Inc., Fremont, CA, USA) [41,44]. The moss-free soils were allowed to equilibrate after manually removing mosses in collars each year, hence resulting in minimal effects of moss removal on soil aeration and compaction. During the snow-free periods each year, soil CO_2_ and CH_4_ fluxes were generally determined twice to three times each month, and in the snow-covered winter season (from December to early March) no measurement was performed due to ground snowpack and low fluxes. The measurement of soil CO_2_ and CH_4_ fluxes and collection of forest throughfall samples in 2020 was not performed, due to the lockdown of COVID-19. Upon each measurement of soil CO_2_ and CH_4_ fluxes, soil volumetric water content (%, *v*/*v*) and temperature inside all soil collars at 7 cm depth were measured using a combined temperature and moisture probe attached to the portable gas analyzer. The atmospheric pressure data upon each flux measurement were recorded on the gas analyzer. Fluxes of CO_2_ and CH_4_ were calculated using linear regressions of headspace gas concentrations versus the closure time, together with Ideal Gas Law, mostly with the determination coefficients of linear regressions (*R*^2^) more than 0.98 across all measurements. Both CO_2_ and CH_4_ fluxes were expressed as μmol C-CO_2_ m^−2^ s^−1^ and nmol C-CH_4_ m^−2^ s^−1^, respectively.

To study the relationships between moss-mediated CO_2_ and CH_4_ fluxes versus the δ^13^C and δ^18^O values of respired CO_2_, static chambers were utilized in autumn seasons of 2019, 2021 and 2024 to collect gas samples released from soil collars, using airtight 100 mL syringes and 300 mL gas bags at regular closure time (e.g., 0, 5, and 10 min) in each experimental plot [45], and a CO_2_ isotope spectrum analyzer (CCIA-38d-EP, Los Research Inc., Fremont, CA, USA) was used within one week after gas sampling to measure CO_2_ concentration in gas samples and its ^13^C/^12^C and ^18^O/^16^O isotopic ratios [41]. Based on Pee Dee Belemnite (PDB) as the standard, the ^13^C/^12^C and ^18^O/^16^O isotopic ratios of gas samples were expressed by δ^13^C and δ^18^O, respectively:δ^13^C or δ^18^O (‰) = (*R*_sample_/*R*_standard_ − 1) × 100,
where *R*_sample_ and *R*_standard_ represent the molar ratios of ^13^C/^12^C or ^18^O/^16^O in sample and standard, respectively.

The δ^13^C or δ^18^O values of CO_2_ released from soil collars without and with moss cover were calculated using the Keeling plot approach [46,47], namely, via the intercept of linear regressions of δ^13^C or δ^18^O values of CO_2_ against the inverse headspace CO_2_ concentrations at various closure time (*n* = 3). The determination coefficients of linear regressions (*R*^2^) were mostly more than 0.85 for the δ^18^O value of respired CO_2_ and more than 0.97 for the δ^13^C value of respired CO_2_, respectively.

### 2.3. Measurements of Soil Properties and DTN Concentration in Throughfall

Soil bulk density at 10 cm depth was measured in autumn seasons of 2019, 2021 and 2024, using the intact soil core method [48]. Fresh soil pH values (soil/water, 1/2.5, *m*/*m*) were determined using a portable pH meter (PB-10, Sartorius, Göttingen, Germany). Soil moisture content was determined gravimetrically by drying soil samples at 105 °C for 48 h. Soil microbial biomass N and C (MBN and MBC, respectively) contents were measured using the chloroform fumigation and extraction method [49]. Concentrations of DTN and DOC in non-fumigated and fumigated soil K_2_SO_4_ extracts and DTN in throughfall samples were measured using a TOC/TN analyzer (Shimadzu TOC-V_CSH_/TN, Tokyo, Japan), and MBN and MBC contents were calculated by the differences in K_2_SO_4_-extractable DTN and DOC contents between fumigated and non-fumigated soils divided by 0.45 [50,51]. To characterize the changes in soil C availability in all N-treated and non-treated plots, UV absorbance of non-fumigated soil K_2_SO_4_ extracts at 254 nm was determined by using a spectrophotometer (Unic 2800A, Shanghai, China) with a 1 cm path-length quartz cell. The special UV absorbance of soil extracts at 254 nm (SUVA_254_) (L mg^−1^ cm^−1^) was calculated by the absorbance of soil extracts at 254 nm divided by the DOC content and multiplied by 100 [52].

### 2.4. Calculation and Statistical Analysis

The daily moss-mediated CH_4_ and CO_2_ fluxes were calculated via the differences in daily CH_4_ and CO_2_ fluxes from soil collars with and without moss cover, assuming minimal effects of long-term moss removal on soil aeration and compaction, and expressed as nmol C-CH_4_ m^−2^ s^−1^ and μmol C-CO_2_ m^−2^ s^−1^, respectively. Here, the impact of CO_2_ produced from CH_4_ oxidation during the collar closure period on the moss-mediated CO_2_ emission was negligible, due to CH_4_ uptake lower three orders of magnitude than CO_2_ emission across all experimental plots. Given the multi-year duration of collar installation, soils were allowed to equilibrate after manually removing moss in collars each year to minimize potential legacy effects. The annual cumulative CH_4_ uptake and CO_2_ emission across all treatments during the snow-free season in 2019–2024 were roughly calculated by summing daily fluxes over time, provided that the means of two adjacent gas sampling were considered daily CH_4_ and CO_2_ fluxes on non-measurement days and that fluxes in winter were negligible [44]. The annual cumulative DTN input via natural throughfall was calculated by summing of DTN concentration multiplied by the throughfall volume at each sampling over the year. Cumulative soil CH_4_ uptake and CO_2_ emission as well as annual throughfall DTN input in 2020 were not included. Annual cumulative CO_2_ emission and CH_4_ uptake with and without moss cover as well as throughfall DTN input under the litter-removal forest floors were expressed as Mg C ha^−1^ yr^−1^, kg C ha^−1^ yr^−1^, and kg N ha^−1^ yr^−1^, respectively. The annual cumulative moss-induced CH_4_ uptake and CO_2_ emission were calculated via the differences in cumulative CH_4_ and CO_2_ fluxes with and without moss cover over the year, respectively. The monthly and annual soil CO_2_ and CH_4_ fluxes, moss-mediated CO_2_ and CH_4_ fluxes, as well as soil moisture and temperature at a depth of 7 cm across all treatments during the five-year experimental period were shown, respectively, with box and box normal plots using OriginPro 2021 (OriginLab Corporation, Northampton, MA, USA).

All measured variables were checked for homogeneity (Levene’s test) and normality (Shapiro–Wilk test) of variance and log-transformed where necessary. A two-way repeated measures analysis of variance (ANOVA) was performed to evaluate the effects of N levels and mosses on the annual cumulative CH_4_ uptake and CO_2_ emission. A univariate repeated measures ANOVA was performed to evaluate the effects of N levels on the annual cumulative moss-mediated CO_2_ and CH_4_ fluxes. The results of ANOVA on the δ^13^C and δ^18^O values of respired CO_2_ and soil properties were shown in Xu et al. [41]. All the data were analyzed by SPSS (version 19.0, IBM Corp., New York, NY, USA).

The contour plots were drawn to show the joint impacts of soil moisture and temperature at a depth of 7 cm on daily soil CO_2_ and CH_4_ fluxes as well as the daily moss-mediated CO_2_ and CH_4_ fluxes, and scatter plots were drawn to show the impacts of annual N input on annual cumulative moss-mediated CO_2_ emission and CH_4_ uptake. Scatter plots were also drawn to show the relationships between annual meteorological factors and the reductions in annual moss-mediated CH_4_ uptake and CO_2_ emission in all N-treated plots relative to the control plots. The above joint or individual effects were fitted with multivariate curve or linear regressions, using OriginPro 2021 (OriginLab Corporation, Northampton, MA, USA). Data on the δ^13^C and δ^18^O values of respired CO_2_ and soil properties in all N-treated and non-treated plots with mosses [41] were used to explore the main influencing factors of moss-mediated daily CO_2_ and CH_4_ fluxes. Pearson correlation analysis in the ‘corrplot’ package was used to assess the correlation among the moss-mediated CO_2_ and CH_4_ fluxes, soil CO_2_ and CH_4_ fluxes, δ^13^C and δ^13^O values of respired CO_2_ as well as soil properties across all experimental plots during the 5-year measurement duration. A random forest analysis was performed to evaluate the variables significantly affecting the moss-mediated CO_2_ and CH_4_ fluxes using the ‘rfPermute’ package. Pearson correlation analysis and random forest analysis were performed using R 4.4.1 [53]. Based on the results of redundancy analysis in Canoco 5.0 (Plant Research International, Wageningen, The Netherlands), the mechanisms involving the daily moss-mediated CO_2_ and CH_4_ fluxes were documented. Based on the results of redundancy analysis and correlation matrices, as well as the established microbial and isotopic processes, N level, MBC, soil pH, soil DTN, CO_2_ and CH_4_ fluxes as well as the δ^13^C values of respired CO_2_ across all experimental plots with moss cover were chosen as predictors to create a priori structural equation modeling (SEM) to assess their direct and indirect effect pathways on the daily moss-mediated CO_2_ and CH_4_ fluxes. The SEM analysis was performed using the software AMOS 24 (SPSS Inc., Chicago, IL, USA), and the overall goodness of fit for the SEM model was evaluated.

## 3. Results

### 3.1. Effects of N Addition and Moss on CH_4_ and CO_2_ Fluxes Across Time Scales

Besides an increase in soil moisture during April to August each year (*p* < 0.05) (Appendix A), the presence of mosses in the control, on average, significantly increased soil moisture at the depth of 7 cm over the 5-year experimental duration (*p* < 0.001) (Appendix A), which was obviously different from the changes in soil moisture caused by mosses across all N-treated plots (Appendix A). Unlike soil moisture, there was generally a relatively small change in moss-mediated soil temperature at the depth of 7 cm across all treatments, with the exception of a significant reduction in the control (*p* < 0.01) (Appendix A).

In the presence of mosses, the average monthly rates of CH_4_ uptake and CO_2_ emission in the control plots over the 5-year experimental duration ranged from 0.268 to 0.953 nmol CH_4_ m^−2^ s^−1^ and from 0.107 to 3.903 μmol CO_2_ m^−2^ s^−1^, respectively (Appendix A), which were significant larger than the rates in all the N-treated experimental plots (*p* < 0.05) (from 0.175 to 0.760 nmol CH_4_ m^−2^ s^−1^ and from 0.147 to 3.198 μmol CO_2_ m^−2^ s^−1^, respectively) (Appendix A). Across all the N-treated and non-treated experimental plots, the presence of mosses significantly decreased average CH_4_ fluxes and increased average CO_2_ fluxes on monthly and daily scales, with the relatively larger differences in the control plots (*p* < 0.05) (Appendix A). Annual cumulative CH_4_ uptake and CO_2_ emission in the control with moss cover ranged from 1.29 to 1.83 kg C ha^−1^ yr^−1^ and from 4.82 to 6.51 Mg C ha^−1^ yr^−1^, respectively, along with the means of 1.60 kg C ha^−1^ yr^−1^ and 5.44 Mg C ha^−1^ yr^−1^, which were larger than those in all N-treated experimental plots, particularly at a high dose (*p* < 0.01) (Table 1). However, in the absence of mosses, no significant differences in the annual cumulative CO_2_ emission and CH_4_ uptake were observed across N-treated and non-treated experimental plots (Table 1). The presence of mosses and N addition and their interaction significantly affected the annual cumulative CO_2_ emission and CH_4_ uptake (*p* ≤ 0.01) (Table 1). Significant interannual differences in the annual cumulative CH_4_ uptake were observed across all treatments, and the stimulating effect of moss cover on annual cumulative CH_4_ uptake varied with years at a marginal significance level (*p* = 0.056) (Table 1).

Regardless of N input, soil moisture and temperature at the depth of 7 cm illustrated the 86–91% variation of daily CO_2_ fluxes and 8–42% variation of daily CH_4_ fluxes across N-treated and non-treated plots with and without moss cover, respectively (Appendix A). A maximum daily CO_2_ flux occurred at soil moisture at the depth of 7 cm ranging from 40% to 60% (*v*/*v*), but it increased with soil temperature ranging from 0 °C to 22 °C (Appendix A). A minimum daily CH_4_ flux occurred at soil moisture at the depth of 7 cm ranging from 15% to 30% (*v*/*v*), along with soil temperature ranging from 2 °C to 8 °C (Appendix A).

### 3.2. Effect of N Addition on Moss-Mediated CH_4_ and CO_2_ Fluxes Across Time Scales

Based on the differences in daily CH_4_ and CO_2_ fluxes in each treatment with and without moss cover, the daily moss-mediated CH_4_ and CO_2_ fluxes across N-treated and non-treated experimental plots had a significant seasonal variation over the 5-year duration, mostly with smaller CH_4_ fluxes in late spring and autumn and larger CO_2_ fluxes in summer (*p* < 0.05) (Figure 1c,d). The seasonal variations in the moss-mediated daily CH_4_ and CO_2_ fluxes were consistent with little rainfall in late spring and autumn and high temperature in summer annually (Figure 1a,b). The relatively smaller daily moss-mediated CH_4_ fluxes and larger daily moss-mediated CO_2_ fluxes occurred in the control plots mostly during the 5-year duration, compared with those across all N-treated experimental plots (*p* < 0.05) (Figure 1c,d). The average monthly rates of moss-mediated CH_4_ fluxes in the control plots over the 5-year duration ranged from −1.66 to −0.33 μmol CH_4_ m^−2^ h^−1^, with a minimum in September, which was much smaller than the rates across all N-treated experimental plots, particularly at a high N dose (*p* < 0.05) (Figure 2a). Except that in March, the average monthly rates of moss-mediated CO_2_ fluxes in the control plots over the 5-year duration were larger than the rates in all N-treated experimental plots (*p* < 0.05), with a maximum difference in August (*p* < 0.01) (Figure 2b). The smallest daily moss-mediated CH_4_ flux and largest daily moss-mediated CO_2_ flux during the 5-year duration was, on average, observed in the control plots, compared with those across all N-treated experimental plots (*p* < 0.001) (Figure 3a,c). The annual accumulative moss-mediated CH_4_ uptake and CO_2_ emission in the control were, on average, 0.77 kg C ha^−1^ yr^−1^ and 2.40 Mg C ha^−1^ yr^−1^, respectively, which were much larger than those in all the N-treated experimental plots, especially at a high N dose (Figure 3b,d).

**Figure 1 plants-15-00166-f001:**
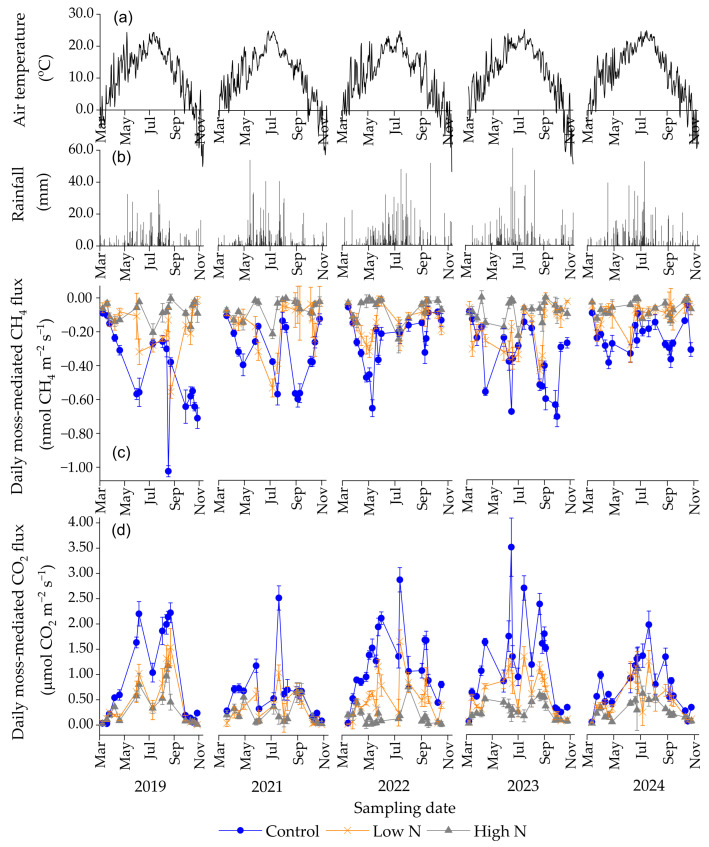
Dynamics of daily mean air temperature (**a**), daily rainfall (**b**), daily moss-mediated CH_4_ (**c**) and CO_2_ (**d**) fluxes from N-treated and non-treated experimental plots from March 2019 to November 2024. In winter (from December to early March) each year, no measurements were taken due to soil freezing and ground snowpack. Bars show standard errors of means (*n* = 4).

The reduction in annual moss-mediated CO_2_ emissions in the high-N-treated plots relative to the control showed an obvious interannual variation, with a maximal reduction in 2022, which differed from that in the low-N-treated plots (Figure 4a). The reduction in annual moss-mediated CH_4_ uptake in all N-treated plots relative to the control reached a maximum in 2019 and then showed a decreasing trend with years, particularly in the high-N-treated plots (Figure 4b), indicating an N-dependent acclimation of moss-mediated CH_4_ uptake. Besides the possible acclimation of moss species or soil microbial communities to chronic N addition, interannual variations in N effects on the annual moss-mediated CO_2_ emission and CH_4_ uptake would be associated with interannual meteorological variables.

**Figure 2 plants-15-00166-f002:**
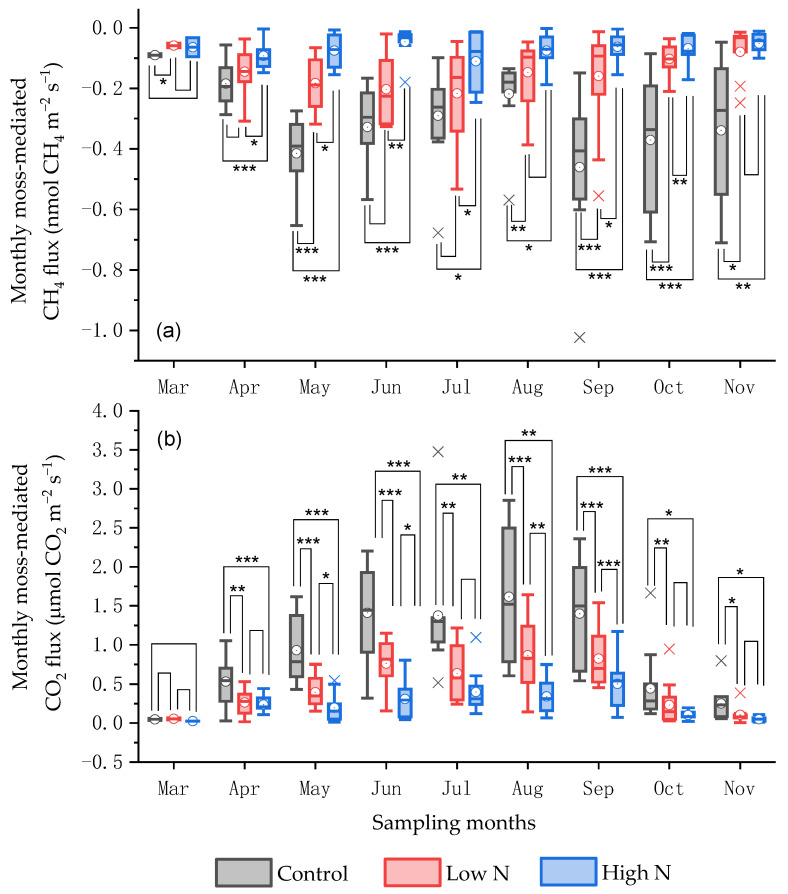
Box plots of monthly moss-mediated CH_4_ (**a**) and CO_2_ (**b**) fluxes from N-treated and non-treated plots from March 2019 to November 2024. Boxes show interquartile (IQR), and circles and horizontal lines in boxes show mean and median values, respectively. Lower and upper whiskers (x) represent 75 percentiles plus 1.5 IQR and 25 percentiles minus 1.5 IQR, respectively. *, *p* < 0.05; **, *p* < 0.01; ***, *p* < 0.001.

**Figure 3 plants-15-00166-f003:**
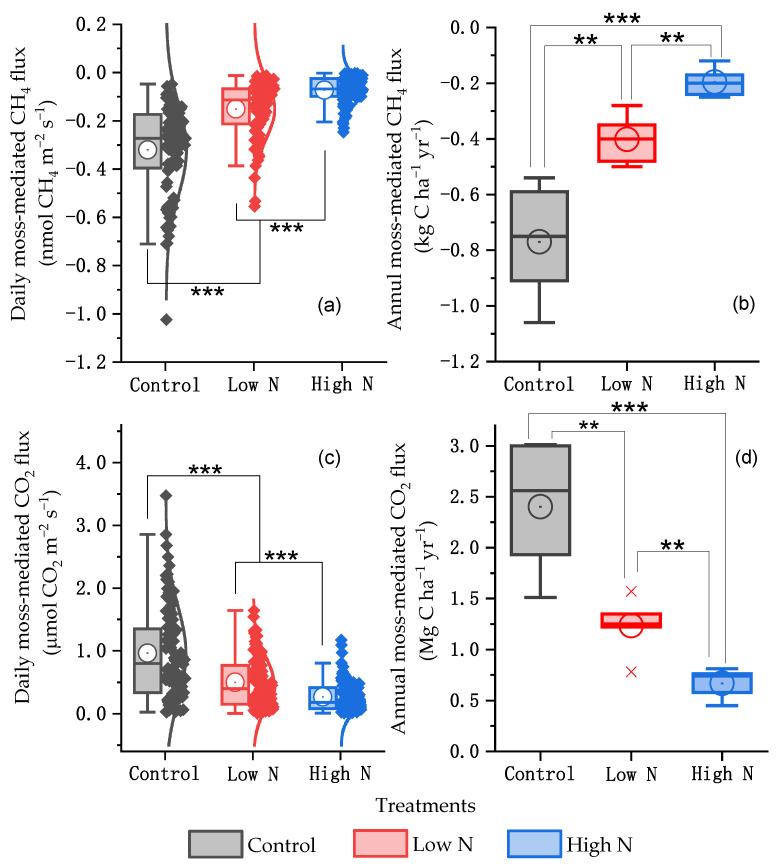
Box normal plots of daily moss-mediated CH_4_ (**a**) and CO_2_ (**c**) fluxes and box plots of annual moss-mediated CH_4_ (**b**) and CO_2_ (**d**) fluxes from N-treated and non-treated plots from March 2019 to November 2024. For the descriptions of the box plots, see the caption of Figure 2. **, *p* < 0.01; ***, *p* < 0.001.

### 3.3. Relationships of Annual Cumulative Moss-Mediated CH_4_ and CO_2_ Fluxes and Meteorological Factors

With an increase of 10 kg N ha^−1^ per year as N input, annual cumulative moss-mediated CH_4_ uptake and CO_2_ emission under the temperate forest was reduced by 0.127 ± 0.019 kg C ha^−1^ yr^−1^ and 0.379 ± 0.062 Mg C ha^−1^ yr^−1^, respectively (Figure 5). The reduction in the annual moss-mediated CO_2_ emissions across all experimentally N-treated plots relative to the control plots became smaller with increasing average annual air temperature (*p* < 0.05) and average non-growing season air temperature (*p* < 0.01), respectively (Figure 6b,c). Furthermore, the reduction in annual moss-mediated CO_2_ emissions caused by the long-term N input was significantly strengthened with the duration of ground snowpack (*p* < 0.01) (Figure 6d). The results strongly indicate that the reduction in snowpack duration and increasing non-growing season air temperature under the context of global warming [54] can, to some extent, weaken the inhibition of increasing atmospheric N deposition on the moss-mediated CO_2_ emission in temperate forests of northeastern China. Contrary to the reduction in annual moss-mediated CO_2_ emission caused by the chronic N input (Figure 6a–d), there were no significant relationships between the reduction in annual moss-mediated CH_4_ uptake caused by the N addition versus average annual air temperature and average annual non-growing season air temperature, and the duration of ground snowpack (Figure 6f–h). However, the inhibition of chronic N input on the annual cumulative moss-mediated CH_4_ uptake under the well-drained temperate forest was significantly strengthened with the reduction of annual rainfall within a range from 600 mm to 1000 mm (*p* < 0.05) (Figure 6e).

**Figure 4 plants-15-00166-f004:**
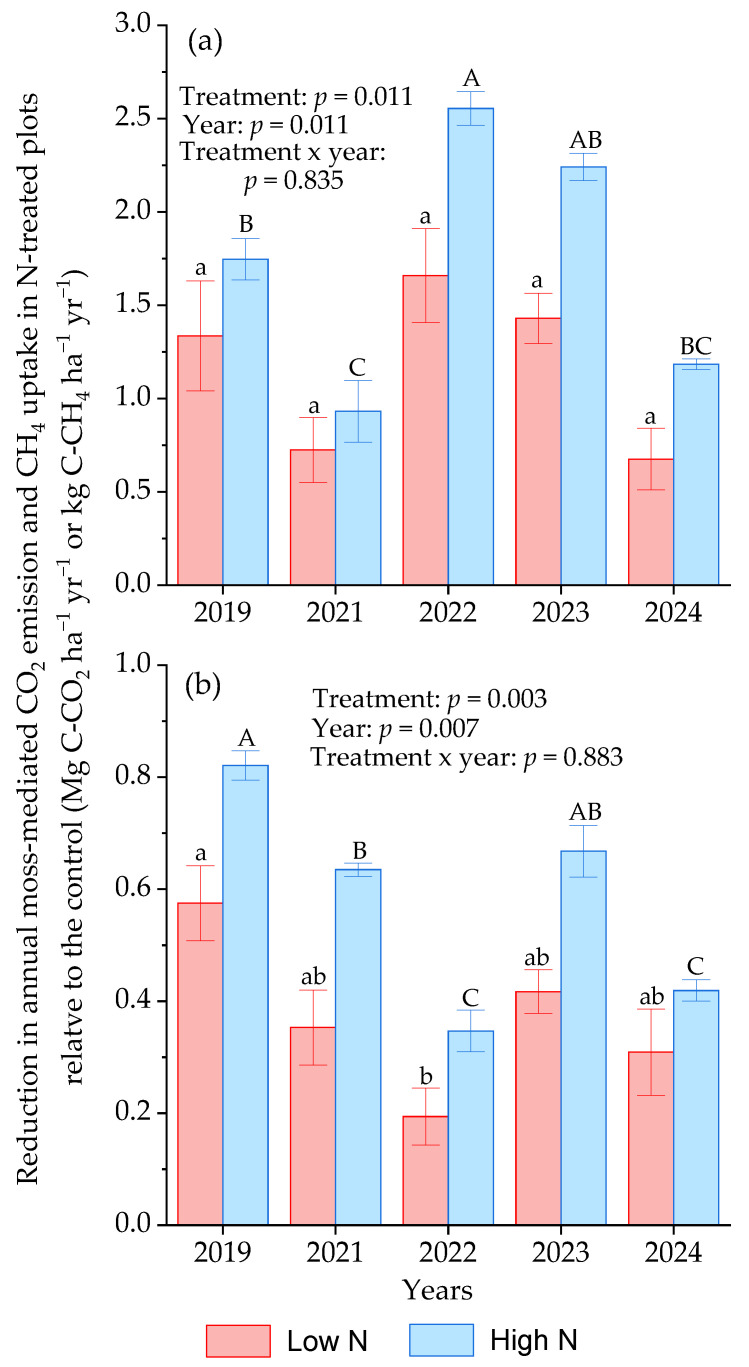
Reduction in annual moss-mediated CO_2_ emission (**a**) and CH_4_ uptake (**b**) in all N-treated plots relative to the control across various years. Bars represent standard errors of means (*n* = 4). Means followed by different small and capital letters represent significant differences in decreased annual moss-mediated C fluxes in the low N and high N-treated plots across various years, respectively. The *p* values of ANOVA analysis are shown.

**Figure 5 plants-15-00166-f005:**
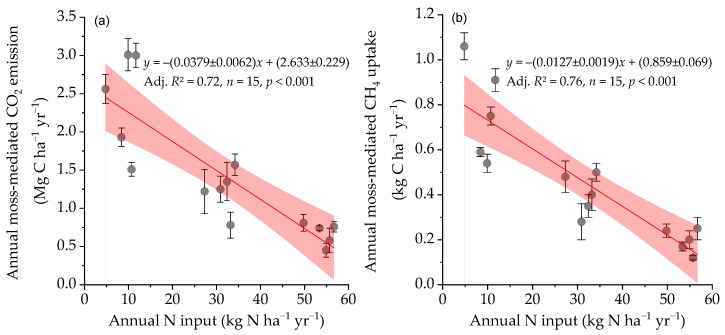
Relationships between annual moss-mediated CO_2_ (**a**) and CH_4_ fluxes (**b**) against the annual N input including natural throughfall over the 5-year experimental duration. Slope and intercept of linear regressions are shown as value ± standard error, and a shaded area represents a 95% confidence band. Adj. *R*^2^, adjusted determination coefficients of linear regressions.

### 3.4. Relationships Between Daily Moss-Mediated CO_2_ and CH_4_ Fluxes and Environmental Variables

Regardless of N addition, soil moisture and temperature at 7 cm depth in the moss-covered plots explained the 7–36% variation in daily moss-mediated CH_4_ fluxes and 24–54% variation in daily moss-mediated CO_2_ fluxes across all treatments, respectively (Appendix A). A minimum daily moss-induced CH_4_ flux occurred at soil moisture ranging from 15% to 30% (*v*/*v*), along with soil temperature ranging from 2 °C to 8 °C (Appendix A). A maximum daily moss-mediated CO_2_ flux was observed at soil moisture ranging from 40% to 60% (*v*/*v*), but it increased with soil temperature ranging from 0 °C to 22 °C (Appendix A). The ranges of soil temperature and moisture causing the extremes of daily moss-mediated CO_2_ and CH_4_ fluxes (Appendix A) were similar to those that resulted in the extremes of daily CH_4_ and CO_2_ fluxes with and without moss cover (Appendix A).

The correlation analysis indicated that daily moss-mediated CO_2_ fluxes in all N-treated and non-treated plots were significantly correlated with soil moisture and temperature, DTN, bulk density, MBC, daily CO_2_ and CH_4_ fluxes as well as the δ^13^C values of respired CO_2_ in the presence of mosses (*p* < 0.05) (Appendix A). Regardless of N input, daily moss-mediated CH_4_ fluxes were negatively correlated with soil pH values (*p* < 0.01) and positively correlated with soil DOC contents (*p* < 0.01) and daily CH_4_ fluxes across the moss-covered plots (*p* < 0.001), respectively (Appendix A). Random forest analysis showed that daily CH_4_ flux, soil moisture and pH values in the moss-covered plots were the strongest predictors of the daily moss-mediated CH_4_ fluxes (*R*^2^ = 0.73, *p* < 0.05) (Appendix A), whereas the daily moss-mediated CO_2_ fluxes were significantly predicted by the daily CO_2_ and CH_4_ fluxes from moss-covered plots (*R*^2^ = 0.62, *p* < 0.05) (Appendix A).

The redundancy analysis further showed significant relationships among the daily moss-mediated CH_4_ and CO_2_ fluxes, N levels, sampling years, and environmental factors in all moss-covered plots (Appendix A). Both axes 1 and 2 explained 100% of total variability in the data obtained under experimental conditions. The control was separated from low N and high N treatments by the Y axis, and low N and high N treatments were distributed in quadrants 2 and 3, respectively. In quadrants 1 and 3 were also distributed three years of soil sampling. The daily moss-mediated CH_4_ and CO_2_ fluxes were distributed in quadrants 1 and 2. The results indicated that both N levels and sampling years affected the daily moss-mediated CH_4_ and CO_2_ fluxes, which in turn correlated closely with the variations in daily CH_4_ and CO_2_ fluxes, δ^13^C and δ^18^C values of respired CO_2_, as well as soil properties in all moss-covered plots. The SEM results showed that the increases in daily moss-mediated CH_4_ fluxes following long-term N addition were associated with the decreases in soil pH values and δ^13^C values of respired CO_2_ and with the increases in MBC and daily CH_4_ flux in all moss-covered plots (Figure 7a,b). The daily moss-mediated CO_2_ fluxes were well regulated by the soil DTN, MBC, δ^13^C values of respired CO_2_, daily CO_2_ fluxes, and N levels in the presence of mosses (Figure 8a,b).

**Figure 6 plants-15-00166-f006:**
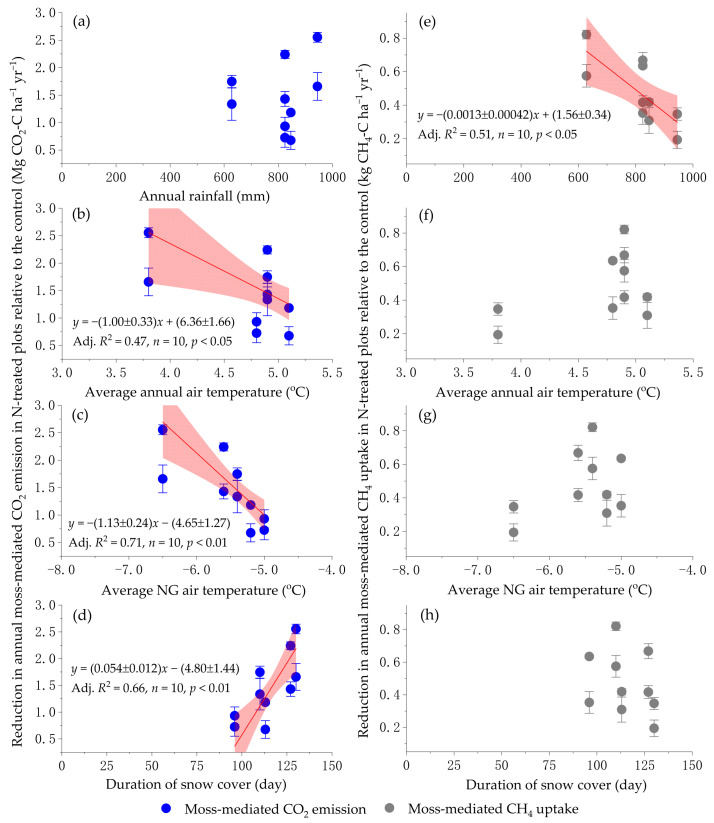
Relationships between the reduction in annual moss-mediated CO_2_ emission (**a**–**d**) and CH_4_ uptake (**e**–**h**) in all experimentally N-treated plots relative to the control against annual meteorological factors. Slope and intercept of linear regressions are shown as value ± standard error, and a shaded area represents a 95% confidence band. Adj. *R*^2^, adjusted determination coefficients of linear regressions; NG, non-growing season (from 1 November to 30 April) at this study site.

**Figure 7 plants-15-00166-f007:**
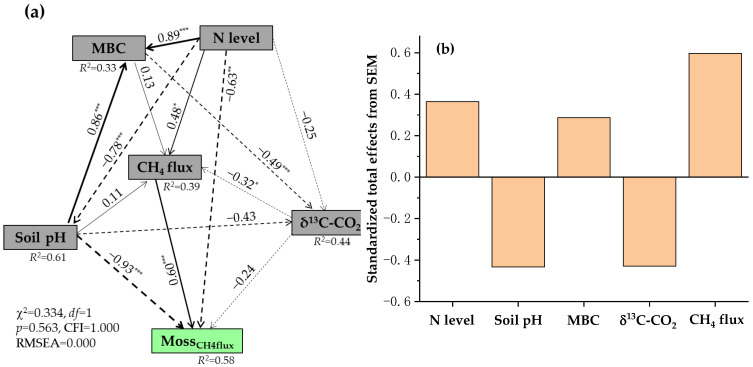
Structural equation modeling (SEM) analysis of causal relationships among N level, soil pH, MBC, δ^13^C-CO_2_, CH_4_ flux in all N-treated and non-treated plots with mosses, and Moss_CH4flux_ (**a**), and their standardized total effects on the Moss_CH4flux_ values from SEM (**b**). Moss_CH4flux_ represents moss-mediated CH_4_ fluxes from N-treated and non-treated experimental plots, respectively. MBC, microbial biomass C. δ^13^C-CO_2_ represents δ^13^C value of carbon dioxide released from N-treated and non-treated plots with mosses. Single-headed arrows indicate the hypothesized direction of causation. Solid and dash arrows indicate positive and negative relationships, respectively. The width of arrows is in proportion to the intensity of the relationship. The numbers near the arrows are the standardized path coefficients, and *R*^2^ values indicate the proportion of variations interpreted by relationships with other variables. χ^2^ represents the Chi-square, CFI for comparative fit index, and RMSEA for root-mean-square error of approximation. *df* and *p* show degrees of freedom and probability level, respectively. *, *p* < 0.05; **, *p* < 0.01; ***, *p* < 0.001.

**Figure 8 plants-15-00166-f008:**
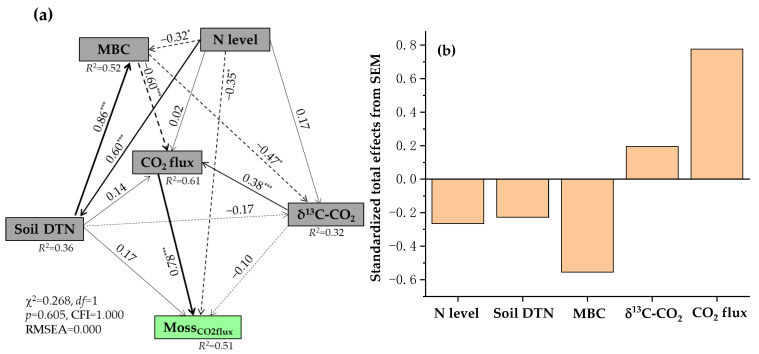
Structural equation modeling (SEM) analysis of causal relationships among N level, soil DTN, MBC, δ^13^C-CO_2_, CO_2_ flux in all N-treated and non-treated plots with mosses, and Moss_CO2flux_ (**a**), and their standardized total effects on the Moss_CO2flux_ values from SEM (**b**). Moss_CO2flux_ represents moss-mediated CO_2_ fluxes from N-treated and non-treated experimental plots, respectively. MBC, microbial biomass C; DTN, dissolved total N. δ^13^C-CO_2_ represents δ^13^C value of carbon dioxide released from N-treated and non-treated plots with mosses. Single-headed arrows indicate the hypothesized direction of causation. Solid and dash arrows indicate positive and negative relationships, respectively. The width of arrows is in proportion to the intensity of the relationship. The numbers near the arrows are the standardized path coefficients, and *R*^2^ values indicate the proportion of variations interpreted by relationships with other variables. χ^2^ represents the Chi-square, CFI for comparative fit index, and RMSEA for root-mean-square error of approximation. *df* and *p* show degrees of freedom and probability level, respectively. *, *p* < 0.05; ***, *p* < 0.001.

## 4. Discussion

### 4.1. Interactive Effects of Mosses and N Addition on Soil CH_4_ Fluxes

The magnitude of the increase in CH_4_ fluxes caused by moss removal became significantly smaller as more N was added to the soil (Appendix A), indicating an interactive effect of mosses and N levels on the soil CH_4_ fluxes under the well-drained temperate forest. The increased soil CH_4_ fluxes upon the removal of mosses were in agreement with the results reported by Chen et al. [9], who showed an increase in CH_4_ fluxes from moss-free forest soil cores under dark and light conditions relative to fluxes from moss-covered soil cores. Furthermore, the maximum increase in the CH_4_ fluxes caused by the moss removal in the control (3.7 μmol C m^−2^ h^−1^) (Figure 1c) was smaller than increases reported by Chen et al. [9] in laboratory incubation studies (6.0–8.0 μmol CH_4_-C m^−2^ h^−1^). Compared with field experimental conditions, laboratory setup (soil cores and controlled moisture 31–41%, *v*/*v*) of Chen et al. [9] facilitated greater atmospheric CH_4_ diffusion into the soil, leading to the larger measured CH_4_ uptake by the moss-covered soil cores [9]. During the 5-year measurement duration, the annual cumulative moss-mediated CH_4_ uptake in the control (Figure 3b) was larger than that reported by Li et al. [10] under a *Pleurozium schreberi* (Brid.) Mitt dominated subalpine coniferous forest floor. This difference is likely due to the subalpine forest site having lower soil pH (4.5), higher annual rainfall (1861 mm), and denser canopy cover (>90%) [10,55] relative to the site-specific forest stand of this study containing a tree canopy cover of 66% [56] and soil pH value of 5.5 [16]. Within a range of soil pH from 4.0 to 7.0, CH_4_ uptake by temperate forest soils decreased with decreasing soil pH [57,58]. Regardless of nutrient fertilization, a larger CH_4_ uptake under *Sphagnum* mosses-dominated boreal forest at a low canopy cover was observed relative to that under *feather* and *acrocarpous* mosses-dominated boreal forest at a dense canopy cover [11]. Contradicting this study, Mason et al. [33] showed that the removal of mosses had no significant effect on CH_4_ fluxes in a boreal fire chronosquence. Therefore, the inhibiting impacts of moss removal on the soil CH_4_ uptake would be variable, mostly depending on site-specific soil properties such as soil moisture, pH, nutrient status (especially N), moss species composition, and forest canopy cover.

The role of moss species (e.g., *Sphagnum* and *Feather* mosses) in regulating CH_4_ fluxes has been documented mostly in poorly drained terrestrial ecosystems (e.g., wetland, peatland, bog, and mire) [14]. Compared with those from *Carex*-dominated fens, CH_4_ emissions from *Sphagnum*-dominated bogs became smaller [59]. A loose symbiosis between methanotrophs and *Sphagnum* species can facilitate the oxidation of CH_4_ by methanotrophic bacteria associated with *Sphagnum*, via the supply of oxygen derived from photosynthesis for methanotrophs and the release of CO_2_ produced by CH_4_ oxidation for photosynthesis of submerged *Sphagnum* mosses [14,60]. However, to date fewer studies have focused on the interactive functions of mosses and chronic N addition in mediating CH_4_ fluxes under well-drained forest floors and the main driving factors [10,11].

The average monthly moss-mediated CH_4_ uptake across all treatments showed a seasonal variation, with the relatively larger values recorded in the control (Figure 2a). Furthermore, the measured forest moss-mediated CH_4_ uptake in the control (0.33 to 1.66 μmol m^−2^ h^−1^) was much smaller than the potential CH_4_ oxidization capacity reported for the moss layer in northern peatlands (20–530 μmol m^−2^ h^−1^) [61]. The high CH_4_ production under waterlogged conditions and oxygen supply via *Sphagnum* photosynthesis can create ideal conditions for symbiotic methanotrophic bacteria, leading to high potential CH_4_ uptake [14,60]. Under well-drained forest floors, lower CH_4_ production and different moss species/conditions result in significantly lower actual CH_4_ uptake. However, annual cumulative moss-mediated CH_4_ uptake in the control plots ranged from 0.54 to 1.06 kg C ha^−1^ yr^−1^, and it was decreased by approximately 50% after a decade of adding low-dose N fertilizer, which was approximately twice the annual DTN input via natural forest throughfall (Figure 3b and Figure 5b). This demonstrates a strong negative influence of long-term, low-level N input on the key CH_4_ sink function of moss species under well-drained temperate forest stands in northeastern China. Given a similar response of CH_4_ flux to the moss removal under various temperate forests in northeastern China containing an estimated forest area of 58.6 million hectares [62] and the inclusion of mosses around the base of tree trunks, it was roughly estimated that mosses would increase annual atmospheric CH_4_ uptake in temperate forest stands of northeastern China by a magnitude of 31.64 to 62.12 Mg C yr^−1^, which was inhibited by the enhanced N deposition. The extrapolation of moss-mediated CH_4_ uptake to the entire temperate forest area of northeastern China was performed by using a first-order approximation, with explicit acknowledgment of variability in moss cover, forest type, and soil conditions. Hence, similar studies should be performed in different climate zones to reasonably evaluate the function of moss species in regulating CH_4_ fluxes from forest ecosystems. Considering that long-term N addition in this study can significantly reduce the annual cumulative CH_4_ uptake mediated by mosses (Figure 3b), and that this inhibitory effect is enhanced with decreasing annual precipitation (Figure 6e); therefore, in the context of climate change, ignoring the roles of moss cover and its sensitivity to increasing atmospheric N deposition in regulating soil CH_4_ fluxes would lead to inaccurate predictions of CH_4_ fluxes in forest ecosystems at regional and global scales.

The moss removal and N addition produced a synergistic inhibition effect on CH_4_ uptake under the temperate forest (*p* < 0.001) (Figure 3b and Table 1). The low dose of added N, along with annual throughfall DTN input in the region (approximately 10.0 kg N ha^−1^ yr^−1^) (Figure 5) would be beyond annual N fixation capacities of moss species mostly reported across various terrestrial ecosystems [63]. Probably the addition of N at low and high doses had harmful effects on the growth of mosses under the experimental conditions. The aboveground biomass of moss species would decrease while it saturated with N [23,25,26,27,32], and the loss of biomass production decreased soil C availability via the leakage of photosynthates [1,8,12]. The decreased soil C availability following the inhibition of moss growth was partly explained by the increased SUVA_254_ values of soil extractable DOC in the high-N-treated experimental plots [41]. The reduction in soil C availability decreases C sources and energy for the activities of soil heterotrophs including methanotrophs [6,7], which may eliminate CH_4_ uptake in well-drained forest soils [16,17,18]. Additionally, pH values of forest soils at 10 cm depth were, on average, reduced by 0.4–0.7 units following the long-term N addition [41], which in turn reduced the soil atmospheric CH_4_ oxidation under well-drained temperate forests [57,58]. Taken together, the decrease in the soil C availability as indicated by the increased SUVA_254_ values [52] and low soil pH following the long-term N input [41] produces a synergistic inhibition of methanotrophs beneath well-drained temperate forest floors mostly covered with mosses (Figure 9), thus leading to a reduction of 48–75% for annual cumulative moss-mediated CH_4_ uptake relative to the control (Figure 3b).

**Figure 9 plants-15-00166-f009:**
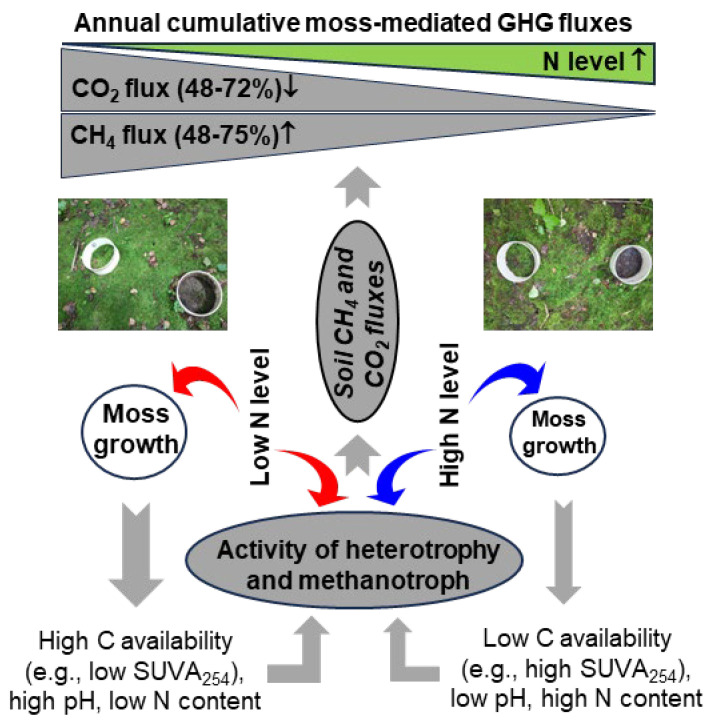
Conceptual scheme for the mechanisms explaining the effects of N levels on annual cumulative moss-mediated CH_4_ and CO_2_ fluxes under well-drained temperate forests. Red arrow, positive effect; blue arrow, negative effect; gray arrow, causal relationships among variables.

### 4.2. Interactive Effects of Mosses and N Addition on Soil CO_2_ Fluxes

Mosses can contribute to organic matter in soil via litter and exudates [1,12] and help buffer soil acidity [8]. Their removal reduces this input, lowering the energy source for soil microbes responsible for both respiration and CH_4_ oxidation. The stimulation of moss growth and its associated biological N fixation (BNF) under low N level conditions can promote the release of labile organic C derived from moss’s photosynthates and BNF into the soil and increases the oxygen tension in moss blankets [6]. The increases in soil labile C and oxygen tension, together with the relatively high soil pH can thus increase the activity of soil heterotrophy and methanotroph (Figure 9). However, the interaction between the enhanced N levels and the decrease in soil pH and C availability upon the long-term high N input will suppress the moss-mediated CO_2_ and CH_4_ fluxes (Figure 9), by integrating the results of correlation matrices (Appendix A), redundancy analysis (Appendix A), and SEM analysis (Figure 7 and Figure 8).

In natural environments, about 50% of the C photosynthetically fixed by plants can be allocated belowground [64,65,66]. The belowground C input would have a potential to increase soil C availability. As a vital plant variety under forest floors, the photosynthetically fixed C by bryophyte species can contribute to 5–50% of gross primary productivity of the tree canopy [3,4], probably affecting soil organic C turnover and CO_2_ flux. Under low N conditions, the growth of mosses was sometimes stimulated [24], and the leakage of photosynthates produced by the moss growth was a good C source for heterotrophic microorganisms active in CO_2_ production [1,8,12]. This could, to some extent, result in higher soil C availability in the control than in the high-N-treated plots, which was partly indicated by the relatively smaller SUVA_254_ values in the control plots [41]. The lower soil C availability in the high-N-treated plots could be ascribed to the negative impacts of long-term, high-level N input on the growth of mosses [24,27]. Salemaa et al. [24] reported that biomass production of three bryophyte species was promoted by the addition of N at less than 2–3 g N m^−2^, but it declined with the high N addition. The results of Salemaa et al. [24] indicated that the N impacts on the growth of mosses would depend on the doses of added N, moss species, and soil moisture. Lu et al. [27] expounded that *Sphagnum* mosses in a temperate peatland could be suppressed following 12-year N addition as 10 kg N ha^−1^ yr^−1^ and even experienced mortality. Due to the low annual natural throughfall DTN input (e.g., approximately 10 kg N ha^−1^ yr^−1^) in the study region (Figure 5), the increased soil C availability likely resulting from the leakage of photosynthates produced by the moss growth would support a larger moss-mediated CO_2_ flux in the control than in all N-treated plots on daily, monthly and annual scales (Figure 2b and Figure 3c,d). Besides the impact of N input on the moss biomass, the growth of mosses increased nonlinearly with ambient temperature [34], which resulted in a maximum of average monthly moss-mediated CO_2_ fluxes across all N-treated and non-treated plots in August annually (Figure 2b), when soil moisture and temperature gave an optimum for the moss-mediated CO_2_ flux (Appendix A). Approximately 51% of the variability of the moss-mediated CO_2_ fluxes was explained by the N levels, soil DTN, MBC, CO_2_ fluxes and δ^13^C values of respired CO_2_ across all moss-covered plots (Figure 8), which indicated that changes in soil N availability and soil organic C turnover in the context of increasing N deposition would contribute to the daily moss-mediated CO_2_ flux (Figure 9). An exceptional finding of this study was that the suppression of long-term N addition on annual cumulative moss-mediated CO_2_ emission significantly reduced with increasing average annual air temperature and average annual non-growing season air temperature as well as with decreasing duration of ground snowpack (Figure 6b–d), indicating that global warming would offset the negative impacts of increasing N deposition on the moss-mediated CO_2_ flux. This was partially ascribed to the thermal acclimation of moss species and tree roots associated with N addition, which needs to be studied under site-specific conditions with varying annual meteorological variables.

Normally, the photosynthates leaked by bryophytes have high contents of carbohydrates, which are generally enriched in ^13^C by 3 to 6‰, relative to lignin and other recalcitrant soil organic matter [35,36,37]. The degradation of photosynthates leaked by mosses may cause a relatively higher enrichment of respired CO_2_ in ^13^C. However, mosses generally depleted ^13^C in respired CO_2_ under the well-drained temperate forest across the three autumn seasons, which was partly modulated by N input [41]. Furthermore, a significant positive correlation between moss-mediated CO_2_ fluxes versus δ^13^C values of respired CO_2_ across all N-treated and non-treated plots with moss cover (*p* < 0.01) (Appendix A) showed that microbial decomposition of soil native organic C inferred from δ^13^C shifts in respired CO_2_ could in turn affect the dynamics of daily moss-mediated CO_2_ fluxes under the experimental conditions. Together with different impacts of soil microbial biomass C and δ^13^C values of respired CO_2_ on daily moss-mediated CO_2_ and CH_4_ fluxes (Appendix A) and no impacts of δ^18^O values of respired CO_2_ on the moss-mediated C fluxes (Appendix A), future studies should focus on the changes in the microbial degradation properties of soil recalcitrant organic C (e.g., the activity of soil microorganisms like *K*-strategist under chronic N-surplus conditions for sustaining microbial nutrient stoichiometry) [67,68,69], functional traits of moss species and tree roots, and soil respiration components (e.g., autotrophic and heterotrophic respiration) and their related isotopic values following chronic increased N input [41,70] under site-specific forests with varying annual meteorological variables, which would affect the dynamics of the moss-mediated CO_2_ and CH_4_ fluxes.

## 5. Conclusions and Implications

Mosses under the well-drained temperate forest floors created a quantifiable, seasonally variable CH_4_ sink (0.54–1.06 kg C ha^−1^ yr^−1^) and CO_2_ emission (1.51–3.01 Mg C ha^−1^ yr^−1^), and the moss-mediated C fluxes became highly sensitive to the chronic N addition, being halved by the long-term, low-level N input under the well-drained temperate forest. The variation in moss-mediated CO_2_ and CH_4_ fluxes upon varying annual N input would result from decreased soil C availability and lowered soil pH as well as the impacts of increased N input on the δ^13^C shifts in CO_2_ released from soil respiration and the growth of mosses. The results of the 5-year field observation indicated that the inhibition of long-term N addition on moss-mediated CO_2_ emission and CH_4_ uptake would vary with different meteorological conditions on annual scales, which needs to be verified under well-drained forest ecosystems in different climate zones. In this study, no measurements were performed to explore the impacts of functional traits of tree roots and moss species on the moss-mediated C fluxes upon increased N input. Future studies would take into account the turnover of soil organic C using isotopic technology, functional traits of moss species and tree roots, and microbial driving mechanisms in the context of increasing N deposition. These findings would have critical implications for explaining and predicting responses of CH_4_ and CO_2_ fluxes at the soil–atmosphere interface in forest ecosystems to increasing N deposition. Accurately predicting how CH_4_ fluxes in forest ecosystems respond to the increasing N deposition necessitates explicitly accounting for the role of moss cover and its interaction with the N enrichment in regulating the soil CH_4_ flux. More field measurements need to be carried out to explore the main driving factors involving the moss-mediated CH_4_ and CO_2_ fluxes under site-specific forests in different climate zones, so that model studies regarding CH_4_ fluxes from forest ecosystems can be performed by reasonably incorporating biotic function of mosses and its responses to the increasing atmospheric N deposition.

## Figures and Tables

**Table 1 plants-15-00166-t001:** Means and ranges of annual CO_2_ emission and annual CH_4_ uptake in N-treated and non-treated plots with and without moss cover, and the results of ANOVA analysis.

Treatments	Cumulative CO_2_ Emission	Cumulative CH_4_ Uptake
	(Mg C ha^−1^ yr^−1^)	(kg C ha^−1^ yr^−1^)
With moss		
Control	5.44 (4.82–6.51)	1.60 (1.29–1.83)
Low N	4.41 (4.13–4.87)	1.48 (1.12–1.79)
High N	4.04 (3.78–4.28)	0.96 (0.76–1.10)
Without moss		
Control	3.04 (2.52–3.51)	0.83 (0.70–1.05)
Low N	3.18 (2.91–3.52)	1.08 (0.84–1.39)
High N	3.37 (2.97–3.79)	0.77 (0.59–0.90)
ANOVA analysis (*p* value)		
Moss	0.001	0.001
Treatment	0.010	0.000
Year	0.116	0.019
Treatment × Year	0.281	0.142
Moss × Year	0.134	0.056
Moss × Treatment	0.000	0.000
Moss × Treatment × Year	0.449	0.714

## Data Availability

The data are contained within the article.

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
