# Peer review of "Inhibition by Nitrogen Addition of Moss-Mediated CH4 Uptake and CO2 Emission Under a Well-Drained Temperate Forest, Northeastern China"

_plants, 2026, doi:10.3390/plants15010166_

Round 1

Reviewer 1 Report

Comments and Suggestions for Authors

Dr Author

This article is based on the white birch forest at the foot of Changbai Mountain in Northeast China. Three treatments were set up: control, low nitrogen (22.5 kg N), and high nitrogen (45 kg N ). After 9 consecutive years of nitrogen application, the CH4 and CO2 fluxes of soil with and without moss cover were monitored from 2019 to 2024 to analyze the impact of nitrogen deposition on moss mediated carbon flux and its driving mechanism. Long term nitrogen addition significantly inhibited moss induced CH4 absorption and CO2 emissions, with a decrease of approximately 48% and 75% under high nitrogen treatment, respectively. Soil pH, available nitrogen, carbon content, and microbial activity (reflected by changes in δ13 C-CO2) are the main driving factors.

  1. Although the main results are mentioned in the abstract, the time span of the study and the core details of the experimental design (such as nitrogen application duration and sampling frequency) are not clearly indicated.Suggest adding a sentence emphasizing the importance of research on forest carbon sink models.
  2. The introduction section is lengthy with numerous references cited, but the logical structure is somewhat loose, with some paragraphs repeatedly emphasizing the relationship between moss and carbon flux.Lack of a clear summary of the 'research gap' and why this study is necessary.
  3. Suggest adding a short concluding paragraph to clearly state the scientific hypotheses and objectives of this study.
  4. The experimental design description is relatively clear, but some key parameters are not explained or not detailed enough:1)It is not specified whether the distribution of moss species in the sample plot is uniform or whether a quantitative coverage survey has been conducted. 2) Not specified how 'moss removal' is operated (physical removal? Is it regularly maintained?). 3)The sampling frequency of δ ¹³ C and δ ¹⁸ O is relatively low (only in autumn), which may affect the explanatory power of interannual variations.
  5. Suggest supplementing soil carbon to nitrogen ratio (C:N) data to enhance the interpretation of carbon availability.
  6. Merge or streamline some charts to highlight key trends.
  7. Always clearly distinguish between "total flux" and "moss induced flux" in the result description to avoid confusion.
  8. Suggest adding a brief analysis of the impact of interannual climate differences (such as rainfall and temperature) on flux.
  9. Increase discussion on research limitations, such as limited sample size, failure to consider differences in moss species, and failure to measure root activity.
  10. Propose more specific future research directions, such as "analyzing the impact of moss species' functional traits on carbon flux" and "conducting similar studies in different climate zones to verify the universality of the conclusions of this study".

Author Response

This article is based on the white birch forest at the foot of Changbai Mountain in Northeast China. Three treatments were set up: control, low nitrogen (22.5 kg N), and high nitrogen (45 kg N). After 9 consecutive years of nitrogen application, the CHand CO2 fluxes of soil with and without moss cover were monitored from 2019 to 2024 to analyze the impact of nitrogen deposition on moss mediated carbon flux and its driving mechanism. Long term nitrogen addition significantly inhibited moss induced CH4 absorption and CO2 emissions, with a decrease of approximately 48% and 75% under high nitrogen treatment, respectively. Soil pH, available nitrogen, carbon content, and microbial activity (reflected by changes in δ13C-CO2) are the main driving factors.

1. Although the main results are mentioned in the abstract, the time span of the study and the core details of the experimental design (such as nitrogen application duration and sampling frequency) are not clearly indicated. Suggest adding a sentence emphasizing the importance of research on forest carbon sink models.

Response: Thanks. additional contents were added to emphasizing the importance of research on forest carbon sink models in the revised manuscript.

2. The introduction section is lengthy with numerous references cited, but the logical structure is somewhat loose, with some paragraphs repeatedly emphasizing the relationship between moss and carbon flux. Lack of a clear summary of the 'research gap' and why this study is necessary.

Response: The introduction section has been refined to explain the importance of this study and unknown problems related to the moss-mediated CH4 and CO2 fluxes under well-drained forest ecosystems.

3. Suggest adding a short concluding paragraph to clearly state the scientific hypotheses and objectives of this study.

Response: The scientific hypotheses and objectives of this study have been refined at the end of the introduction section. In this study, we hypothesize that long-term N enrichment can suppress moss-mediated CH4 uptake and CO2 emission via its negative impacts on moss growth and soil microbial processes under well-drained forests, and that the extent to which increased N input inhibits moss-mediated CH4 uptake and CO2 emission depends on meteorological conditions. The objectives of this study are to explore the effects of long-term N input on the moss-mediated CO2 and CH4 fluxes under the well-drained temperate deciduous forest, northeastern China based on a 5-year field measurement, and the key influencing factors, by taking into account changes in soil properties such as C availability, pH, and δ13C and δ18O values of respired CO2 as well as meteorological factors.

4. The experimental design description is relatively clear, but some key parameters are not explained or not detailed enough:1) It is not specified whether the distribution of moss species in the sample plot is uniform or whether a quantitative coverage survey has been conducted. 2) Not specified how 'moss removal' is operated (physical removal? Is it regularly maintained?). 3) The sampling frequency of δ¹³C and δ¹⁸O is relatively low (only in autumn), which may affect the explanatory power of interannual variations.

Response: Thanks, due to nice comments. In this study, the distribution of moss species across all experimental plots is uniform, and the coverage of mosses in N-treated experimental plots, especially in the high-N-treated plots appeared an obvious reduction. The moss-free soils were allowed to equilibrate after manually removing mosses in collars each year, hence resulting in minimal effects of moss removal on soil aeration and compaction. The measurement of δ¹³C and δ¹⁸O values of respired CO2 was done in autumn seasons of 2019, 2021, and 2024, respectively. These variables were used to explore the changes in the moss-mediated CO2 and CH4 fluxes under N-treated and non-treated experimental plots. Regarding soil sampling three times, there were different hydrothermal conditions and other soil properties in three autumn seasons, which would be reasonable to explore the driving factors of moss-mediated C fluxes under well-drained forest ecosystems in the context of long-term N addition.

5. Suggest supplementing soil carbon to nitrogen ratio (C:N) data to enhance the interpretation of carbon availability.

Response: In this study, the contents of soil organic C and total N were not determined across all experimental plots following the 9-year N addition. However, the changes in special UV absorbance of soil extracts at 254 nm can reflect soil carbon availability (Xu et al., 2012), which is nicely used to explain the inhibition of long-term N addition on the moss-mediated CO2 emission and CH4 uptake under well-drained temperate forests.

Xu, X.K.; Luo, X.B.; Jiang, S.H.; Xu, Z.J. Biodegradation of dissolved organic carbon in soil extracts and leachates from a temperate forest stand and its relationship to ultraviolet absorbance. Chinese Science Bulletin 2012, 57, 912–920.

6. Merge or streamline some charts to highlight key trends.

Response: The text has been refined to highlight key findings of this study and limitations for future studies.

7. Always clearly distinguish between "total flux" and "moss induced flux" in the result description to avoid confusion.

Response: The text was refined to reduce the confusion of rephrases as much as possible.

8. Suggest adding a brief analysis of the impact of interannual climate differences (such as rainfall and temperature) on flux.

Response: Thanks, due to nice comments. In the revised manuscript, some meteorological variables (e.g., average annual air temperature and average non-growing season air temperature, annual rainfall, and the duration of ground snowpack) were used to explore their relationships with the reduction of annual moss-mediated CO2 emission and CH4 uptake across all N-treated experimental plots relative to the control plots. The corresponding contents were added in the section 3.3. A new figure was nicely added in the text and shown as Figure 6. Considering the relationships of annual cumulative moss-mediated CH4 and CO2 fluxes and annual meteorological factors, the additional objective of this study was taken into account at the end of the introduction section - the extent to which increased N input inhibits moss-mediated CH4 uptake and CO2 emission depends on meteorological conditions.

9. Increase discussion on research limitations, such as limited sample size, failure to consider differences in moss species, and failure to measure root activity.

Response: These contents were mentioned in the discussion section and conclusions and implications section, respectively.

10. Propose more specific future research directions, such as "analyzing the impact of moss species' functional traits on carbon flux" and "conducting similar studies in different climate zones to verify the universality of the conclusions of this study".

Response: Thanks, due to nice suggestion. These contents were added in the discussion section and conclusions and implications section, respectively.

Reviewer 2 Report

Comments and Suggestions for Authors

Brief summary. This manuscript investigates how long-term nitrogen (N) addition alters moss-mediated methane (CH₄) uptake and carbon dioxide (CO₂) emission in a well-drained temperate forest in northeastern China, using a rare combination of a >9-year N manipulation experiment and 5 years of intensive gas flux measurements. The study provides quantitative evidence that increasing N input significantly suppresses moss-induced CH₄ uptake and CO₂ emission, and integrates isotopic (δ¹³C, δ¹⁸O), soil physicochemical, and multivariate/SEM analyses to identify mechanistic drivers. Its main strengths are the exceptional experimental duration, the paired moss/no-moss design, and the attempt to link biogeochemical fluxes with microbial and isotopic indicators, which together make a valuable contribution to forest C-cycle modeling under rising N deposition.

General comments. The manuscript is clearly written, well structured, and highly relevant to forest biogeochemistry and global change research. The research question is well motivated, and the hypothesis that long-term nitrogen enrichment suppresses moss-mediated carbon fluxes is testable and largely supported by the data. A key strength of the study is the paired design with and without moss cover, which allows flux differences to be attributed specifically to moss effects.

The experimental approach is generally appropriate, but some conceptual and methodological aspects need clarification. In particular, interpreting “moss-induced” fluxes solely as differences between collars assumes minimal long-term effects of moss removal on soil structure, microbial communities, and gas diffusion. This assumption should be more clearly justified or explicitly acknowledged as a limitation. In addition, although the absence of winter measurements is understandable, the assumption that winter fluxes are negligible should be supported by relevant literature or brief site-specific evidence, as cold-season CH₄ uptake can be significant in temperate and boreal forests.

The statistical analyses are sound overall, but clearer reporting of effect sizes and associated uncertainty would strengthen key results, especially the regressions linking nitrogen input to cumulative moss-induced fluxes. The SEM analysis is a valuable component of the study; however, the causal structure of the model would benefit from stronger justification based on established microbial and isotopic processes, to avoid over-interpreting correlative relationships.

The conclusions are generally consistent with the results, although regional-scale extrapolations of CH₄ uptake should be presented more cautiously. Ethics and data availability are not a concern for this field study, but a more explicit statement on data accessibility would improve transparency.

Specific comments

The hypothesis that long-term N input and moss presence “synergistically affect” CH₄ and CO₂ fluxes is conceptually sound, but the term synergistic implies a non-additive interaction. While interaction terms are tested statistically, the ecological mechanism underlying this synergy should be articulated more explicitly, particularly how N alters moss physiology versus soil microbial processes. (lines 153–160)

The calculation of moss-induced fluxes as the difference between moss-covered and moss-free collars assumes that moss removal does not alter soil aeration, compaction, or microbial community composition beyond moss effects. Given the multi-year duration of collar installation, please clarify whether soils were allowed to equilibrate after moss removal and discuss potential legacy effects. (257–261)

The assumption that winter CO₂ and CH₄ fluxes are negligible should be supported with references or site-specific evidence. Even a short justification citing comparable temperate forest studies would strengthen confidence in annual budget estimates. (265–266)

The use of δ¹³C-CO₂ as an indicator of labile C inputs and microbial turnover is appropriate, but alternative explanations (e.g., shifts in autotrophic vs. heterotrophic respiration, diffusion effects under moss cover) should be more explicitly acknowledged. The Discussion currently leans toward a single dominant interpretation. Isotopic interpretation (lines 431–435; Figures 5, 6)

While SEM results are statistically convincing, the manuscript should clarify why certain variables (e.g., soil pH, MBC, δ¹³C-CO₂) are treated as mediators rather than parallel responses to N addition. A brief justification of path selection based on prior studies would reduce the risk of over-interpreting causal direction. (Figures 5 and 6)

The linear relationships between annual N input and moss-induced fluxes are central to the manuscript. Please report confidence intervals for regression slopes in the text to better convey uncertainty and robustness. (Figure 4)

The extrapolation of moss-mediated CH₄ uptake to the entire temperate forest area of northeastern China is interesting but potentially overreaching. This estimate should be clearly labeled as a first-order approximation, with explicit acknowledgment of variability in moss cover, forest type, and soil conditions. Regional extrapolation ( lines 528–534)

Several important results (e.g., correlation matrices, random forest analysis, RDA) are confined to the Supplementary Material. The main text would benefit from clearer cross-referencing and a short interpretive summary explaining how these analyses complement the core findings

additional questions for the authors

Did the species composition, thickness, or biomass of the moss layer change over the 9+ years of nitrogen addition?

Given that measurements were conducted under field light conditions, to what extent could seasonal or interannual variability in moss photosynthetic activity (e.g., light availability, hydration state) modulate the magnitude of moss-induced CH₄ and CO₂ fluxes, and how might this interact with nitrogen addition?

Is there evidence that mosses or soil microbial communities acclimated to chronic N addition over time, for example through a weakening or strengthening of N effects in later years of the experiment? If not explicitly tested, could this be explored using the existing time-series data?

The relationships between annual N input and moss-induced fluxes are modeled as linear. Did the authors explore potential nonlinear responses or threshold effects, particularly between low and high N treatments, that could indicate ecosystem tipping points?

Although winter fluxes were assumed negligible, could episodic thaw events or subnivean gas exchange contribute meaningfully to annual CH₄ or CO₂ budgets at this site, particularly under future climate warming scenarios?

Author Response

Brief summary. This manuscript investigates how long-term nitrogen (N) addition alters moss-mediated methane (CH₄) uptake and carbon dioxide (CO₂) emission in a well-drained temperate forest in northeastern China, using a rare combination of a >9-year N manipulation experiment and 5 years of intensive gas flux measurements. The study provides quantitative evidence that increasing N input significantly suppresses moss-induced CH₄ uptake and CO₂ emission, and integrates isotopic (δ¹³C, δ¹⁸O), soil physicochemical, and multivariate/SEM analyses to identify mechanistic drivers. Its main strengths are the exceptional experimental duration, the paired moss/no-moss design, and the attempt to link biogeochemical fluxes with microbial and isotopic indicators, which together make a valuable contribution to forest C-cycle modeling under rising N deposition.

Response: Thanks, due to the reviewer’ encouragement.

General comments. The manuscript is clearly written, well structured, and highly relevant to forest biogeochemistry and global change research. The research question is well motivated, and the hypothesis that long-term nitrogen enrichment suppresses moss-mediated carbon fluxes is testable and largely supported by the data. A key strength of the study is the paired design with and without moss cover, which allows flux differences to be attributed specifically to moss effects.

Response: Thanks. As mentioned above, this is a nice hypothesis that long-term nitrogen enrichment suppresses moss-mediated carbon fluxes under well-drained forest stands, which can be nicely supported by the experimental data.

The experimental approach is generally appropriate, but some conceptual and methodological aspects need clarification. In particular, interpreting “moss-induced” fluxes solely as differences between collars assumes minimal long-term effects of moss removal on soil structure, microbial communities, and gas diffusion. This assumption should be more clearly justified or explicitly acknowledged as a limitation. In addition, although the absence of winter measurements is understandable, the assumption that winter fluxes are negligible should be supported by relevant literature or brief site-specific evidence, as cold-season CH₄ uptake can be significant in temperate and boreal forests.

Response: Thanks, due to this nice comment. In the revised manuscript, the words like “moss-induced” fluxes were changed into “moss-mediated” fluxes. The above-mentioned assumptions have been incorporated into the M&M section. Additional reference (Xu et al., 2023) was added to support that winter carbon fluxes are negligible under site-specific conditions due to soil freezing.

Xu XK, Xu TT, Yue J (2023) Effect of in situ large soil column translocation on CO2 and CH4 fluxes under two temperate forests of northeastern China. Forests, 14, 1531. https://doi.org/10.3390/f14081531

The statistical analyses are sound overall, but clearer reporting of effect sizes and associated uncertainty would strengthen key results, especially the regressions linking nitrogen input to cumulative moss-induced fluxes. The SEM analysis is a valuable component of the study; however, the causal structure of the model would benefit from stronger justification based on established microbial and isotopic processes, to avoid over-interpreting correlative relationships.

Response: Thanks, due to the nice comments. Regarding the linear regressions linking annual nitrogen input to cumulative moss-mediated CH4 uptake and CO2 emission, the standard errors for the slopes of linear regressions were mentioned in the text. A shade area shown in Figure 4 represents a 95% confidence band. Regarding the SEM analysis, the causal structure of the model would be made, by considering the results of redundancy analysis and correlation matrices, the established microbial and isotopic processes, as well as the overall goodness of fit.

The conclusions are generally consistent with the results, although regional-scale extrapolations of CH₄ uptake should be presented more cautiously. Ethics and data availability are not a concern for this field study, but a more explicit statement on data accessibility would improve transparency.

Response: Thanks, due to the nice comments. The data are contained within the article, including supplementary file. The datasets in this study are available from the corresponding author on reasonable request.

Specific comments

The hypothesis that long-term N input and moss presence “synergistically affect” CH₄ and CO₂ fluxes is conceptually sound, but the term synergistic implies a non-additive interaction. While interaction terms are tested statistically, the ecological mechanism underlying this synergy should be articulated more explicitly, particularly how N alters moss physiology versus soil microbial processes. (lines 153–160)

Response: Thanks, due to the nice comments. In the revised manuscript, the words like “synergistically affect” were changed into “interactively affect”, because the absence of mosses and long-term N addition would have negative effects on CO2 and CH4 fluxes under well-drained forest ecosystems. The hypotheses of this study were refined at the end of the introduction section, by considering nice comments of anther reviewer, which are shown at the bottom.

In the context of increasing atmospheric N deposition, we hypothesize that long-term N enrichment can suppress moss-mediated CH4 uptake and CO2 emission via its negative impacts on moss growth and soil microbial processes under well-drained forests, and that the extent to which increased N input inhibits moss-mediated CH4 uptake and CO2 emission depends on meteorological conditions.

The calculation of moss-induced fluxes as the difference between moss-covered and moss-free collars assumes that moss removal does not alter soil aeration, compaction, or microbial community composition beyond moss effects. Given the multi-year duration of collar installation, please clarify whether soils were allowed to equilibrate after moss removal and discuss potential legacy effects. (257–261)

Response: Thanks, due to the nice comments. In this study, the moss-free soils were allowed to equilibrate after manually removing mosses in collars each year, hence resulting in minimal effects of moss removal on soil aeration and compaction. The related text and a corresponding assumption were mentioned in the M&M section of the revised manuscript.

The assumption that winter CO₂ and CH₄ fluxes are negligible should be supported with references or site-specific evidence. Even a short justification citing comparable temperate forest studies would strengthen confidence in annual budget estimates. (265–266)

Response: Here, one reference was nicely cited to explain that winter CO₂ and CH₄ fluxes are negligible under site-specific conditions due to soil freezing, which was mentioned previously.

The use of δ¹³C-CO₂ as an indicator of labile C inputs and microbial turnover is appropriate, but alternative explanations (e.g., shifts in autotrophic vs. heterotrophic respiration, diffusion effects under moss cover) should be more explicitly acknowledged. The Discussion currently leans toward a single dominant interpretation. Isotopic interpretation (lines 431–435; Figures 5, 6)

Response: Normally, soil autotrophic versus heterotrophic respiration and diffusion effects under moss cover would affect the dynamics of isotopic (δ¹³C, δ¹⁸O) values of respired CO2, which was reported in previous reference (Xu et al., 2025). In this study, no measurements of isotopic (δ¹³C, δ¹⁸O) values of respired CO2 from autotrophic and heterotrophic respiration, because bryophytes contain no roots. Due to the addition of two new figures, Figures 5 and 6 in the original version of this manuscript were changed into Figures 7 and 8 in the revised manuscript, respectively.

Xu XK, Kong YH, Feng EP, Yue J, Cheng WG, Khoroshaev D, Kivalov S (2025) Effect of mosses and long-term N addition on δ13C and δ18O values of respired CO2 under a temperate forest floor. Plants, 14, 2707. https://doi.org/10.3390/plants14172707

While SEM results are statistically convincing, the manuscript should clarify why certain variables (e.g., soil pH, MBC, δ¹³C-CO₂) are treated as mediators rather than parallel responses to N addition. A brief justification of path selection based on prior studies would reduce the risk of over-interpreting causal direction. (Figures 5 and 6)

Response: Based on the results of redundancy analysis and correlation matrices, as well as the established microbial and isotopic processes, N level, MBC, soil pH, soil DTN, CO2 and CH4 fluxes as well as the δ13C values of respired CO2 across all experimental plots with moss cover were chosen as predictors to create a priori structural equation modeling (SEM) to assess their direct and indirect effect pathways on the daily moss-mediated CO2 and CH4 fluxes. There were no measurements of soil properties without moss cover, and parallel responses to N addition cannot include N levels. Thus, in this study were not used the parallel responses to N addition, which are normally used in some meta-analysis for integrating various study cases.

The linear relationships between annual N input and moss-induced fluxes are central to the manuscript. Please report confidence intervals for regression slopes in the text to better convey uncertainty and robustness. (Figure 4)

Response: Thanks. Figure 4 in the original version of this manuscript was changed into Figures 5 in the revised manuscript. A shade area shown in Figure 5 represents a 95% prediction band. In the text, standard errors of regression slopes were reported in the text.

The extrapolation of moss-mediated CH₄ uptake to the entire temperate forest area of northeastern China is interesting but potentially overreaching. This estimate should be clearly labeled as a first-order approximation, with explicit acknowledgment of variability in moss cover, forest type, and soil conditions. Regional extrapolation (lines 528–534)

Response: Thanks, due to the nice comments. The first-order approximation and related limitations were mentioned in the text, which was shown at the bottom.

The extrapolation of moss-mediated CH4 uptake to the entire temperate forest area of northeastern China was done by using a first-order approximation, with explicit acknowledgment of variability in moss cover, forest type, and soil conditions. Hence, similar studies should be done in different climate zones to reasonably evaluate the function of moss species in regulating CH4 fluxes from forest ecosystems. Considering that long-term N addition in this study can significantly reduce annual cumulative moss-mediated CH4 uptake (Figures 3b), and this suppression is enhanced with decreasing annual precipitation (Figures 6e), ignoring the roles of moss cover and its sensitivity to increasing atmospheric N deposition in regulating the soil CH4 fluxes would lead to inaccurate predictions of CH₄ fluxes from forest ecosystems at regional and global scales in the context of climate change.

Several important results (e.g., correlation matrices, random forest analysis, RDA) are confined to the Supplementary Material. The main text would benefit from clearer cross-referencing and a short interpretive summary explaining how these analyses complement the core findings.

Response: Based on the results of redundancy analysis and correlation matrices, as well as the established microbial and isotopic processes, N level, MBC, soil pH, soil DTN, CO2 and CH4 fluxes as well as the δ13C values of respired CO2 across all experimental plots with moss cover were chosen as predictors to create a priori structural equation modeling (SEM) to assess their direct and indirect effect pathways on the daily moss-mediated CO2 and CH4 fluxes. The interaction between the enhanced N levels and the decrease in soil pH and C availability upon the long-term high N input will suppress the moss-mediated CO2 and CH4 fluxes (Figure 9), by integrating the results of correlation, RDA, and SEM analysis.

additional questions for the authors

Did the species composition, thickness, or biomass of the moss layer change over the 9+ years of nitrogen addition?

Response: The distribution of moss species across all experimental plots was uniform, and the coverage of mosses in long-term N-treated experimental plots, especially in the high-N-treated plots appeared an obvious reduction. In this study we did not measure the changes in species composition over the 9+ years of nitrogen addition.

Given that measurements were conducted under field light conditions, to what extent could seasonal or interannual variability in moss photosynthetic activity (e.g., light availability, hydration state) modulate the magnitude of moss-induced CH₄ and CO₂ fluxes, and how might this interact with nitrogen addition?

Response: In this study, CO2 and CH4 fluxes from moss-covered and moss-free soil collars were measured on non-rainy days at regular intervals during the snow cover-free periods annually, using a portable greenhouse gas analyzers (915-0011, Los Research Inc., Fremont, CA, USA) coupled to an opaque smart respiratory chamber (SC-11, Beijing LICA United Technology Limited, Beijing, China) (Xu et al., 2023, 2025). Thus, we cannot explore that the seasonal or interannual variability in moss photosynthetic activity (e.g., light availability, hydration state) and its relationships with the moss-induced CH4 and CO2 fluxes under long-term N addition. Anyway, this would be an interesting study. Based on the negative correlation between annual rainfall and the reduction of annual moss-mediated CH4 uptake caused by the long-term N input, it can be concluded that the inhibition of long-term N input on the moss-mediated CH4 uptake under the well-drained temperate forest was significantly strengthened with the reduction of annual rainfall within a range from 600 to 1000 mm (p < 0.05) (Figure 6e). Furthermore, the results indicated that the reduction of snowpack duration and increasing non-growing season air temperature under the context of global warming can, to some extent, weaken the inhibition of increasing atmospheric N deposition on the moss-mediated CO2 emission (Figure 6b-d). Together with these results, the extent to which increased N input inhibits moss-mediated CH4 uptake and CO2 emission can depend on annual meteorological conditions, which can verify one hypothesis mentioned at the end of the introduction section.

Xu XK, Xu TT, Yue J (2023) Effect of in situ large soil column translocation on CO2 and CH4 fluxes under two temperate forests of northeastern China. Forests, 14, 1531. https://doi.org/10.3390/f14081531

Xu XK, Kong YH, Feng EP, Yue J, Cheng WG, Khoroshaev D, Kivalov S (2025) Effect of mosses and long-term N addition on δ13C and δ18O values of respired CO2 under a temperate forest floor. Plants, 14, 2707. https://doi.org/10.3390/plants14172707

Is there evidence that mosses or soil microbial communities acclimated to chronic N addition over time, for example through a weakening or strengthening of N effects in later years of the experiment? If not explicitly tested, could this be explored using the existing time-series data?

Response: In this study, there were no direct measurements for mosses species or soil microbial communities under chronic N addition. Using the existing time-series data, we look at an obviously weakening of N effects on the annual moss-mediated CH4 uptake in later years of the experiment (Figure 4b), indicating an acclimation of moss-mediated CH4 uptake to chronic N addition. Furthermore, the reduction of annual moss-mediated CO2 emissions in the high-N-treated plots relative to the control showed an obvious interannual variation, with a maximal reduction in 2022, which differed from that in the low-N-treated plots (Figure 4a). Besides the acclimation of moss species or soil microbial communities to chronic N addition, interannual variations of N effects on both annual moss-mediated CO2 emissions and CH4 uptake would be mainly associated with interannual meteorological variable (e.g., average annual air temperature and average non-growing season air temperature, the duration of ground snowpack, and annual rainfall), which was partly explained in Figure 6.

The relationships between annual N input and moss-induced fluxes are modeled as linear. Did the authors explore potential nonlinear responses or threshold effects, particularly between low and high N treatments, that could indicate ecosystem tipping points?

Response: The relationships between annual N input and annual cumulative moss-mediated fluxes are modeled as linear, with a significant level. Upon checking potential nonlinear responses or threshold effects, we did not observe the ecosystem tipping points in this study, due to limited dosages of added N across all years.

Although winter fluxes were assumed negligible, could episodic thaw events or subnivean gas exchange contribute meaningfully to annual CH₄ or CO₂ budgets at this site, particularly under future climate warming scenarios?

Response: I agree with your hypothesis that episodic thaw events or subnivean gas exchange would contribute meaningfully to annual CH₄ or CO₂ budgets. In this study, CO2 and CH4 fluxes in all experimental plots were measured during the period from March to November each year, which include episodic thaw events in autumn and late spring season at this site. In winter (from December to early March) each year, no measurements of CO2 and CH4 fluxes in the field were taken due to soil freezing and ground snowpack, and winter fluxes become negligible (Xu et al., 2023).

Reviewer 3 Report

Comments and Suggestions for Authors

The work is interesting. It is edited relatively well. The comments are mainly editorial. The biggest mistake concerns the references. Items from the literature should be listed in the order in which they are cited, not alphabetically. The text of the references should also be edited correctly, in accordance with the editorial requirements. 

Author Response

The work is interesting. It is edited relatively well. The comments are mainly editorial. The biggest mistake concerns the references. Items from the literature should be listed in the order in which they are cited, not alphabetically. The text of the references should also be edited correctly, in accordance with the editorial requirements. 

Response: In the revised manuscript, the reference section and the items of citation in the text were revised, in accordance with the editorial requirements.